



# Coherency and time lag analyses between MODIS vegetation indices and climate across forest and grasslands in European temperate zone

Kinga Kulesza[1], Agata Hościło[1]

[1]Centre of Applied Geomatics, Institute of Geodesy and Cartography, 27 Modzelewskiego Street, 02-679 Warsaw, Poland

*Correspondence to*: Kinga Kulesza (kinga.kulesza@igik.edu.pl)

**Abstract:** Identifying the climate-induced variability in the condition of vegetation is particularly important in the context of the recent climate change, and plants' impact on mitigation of the climate change. In this paper, we present the coherence and time lags in the spectral response of three individual vegetation types in European temperate zone to the influencing meteorological factors, in the period 2002-2022. Vegetation condition in broadleaved forest, coniferous forest and pastures

was measured with monthly anomalies of two spectral indices – NDVI and EVI. As meteorological elements we used monthly anomalies of temperature (T), precipitation (P), vapour pressure deficit (VPD), evapotranspiration (ETo), and teleconnection indices North Atlantic Oscillation (NAO) and North Sea Caspian Pattern (NCP). Periodicity in the time series was assessed using the Wavelet Transform, but no significant intra- or interannual cycles were detected in both vegetation (NDVI and EVI) and meteorological variables. In turn, coherence between NDVI/EVI and meteorological elements was described using the

methods of Wavelet Coherence and Pearson's linear correlation with time lag. In European temperate zone analysed in this study, NAO produces strong coherence mostly for forests, with circa 1 year delay and – a weaker coherence – with circa 3 year delay. For pastures these interannual patterns are hardly recognizable. The strongest relationships occur between condition of the vegetation and T and ETo – they show high coherence in both forests and pastures. There is a significant cohesion with 8-16 month (ca. 1 year) delay and 20-32 month (ca. 2 year) delay. More time lagged significant correlations between vegetation

indices and T occur for forests than for pastures, suggesting a significant lag in the forests' response to the changes in T.

## 1 Introduction

Vegetation is one of the main components of the terrestrial Earth, which plays an important role in regulating climate, through evaporative cooling processes and carbon sequestration, among others. Hence, vegetation's presence between the atmosphere, hydrosphere and lithosphere is crucial (Zhang et al., 2017). Among different vegetation types, the major ones, which cover up

to 78% of the world's land area, are forests and grasslands (Ipcc, 2019; Fao and Unep, 2020).

Modern climate change is widespread, rapid and intensifying (Ipcc, 2019). Climate change deepens the processes of land degradation through e.g. increase in rainfall intensity and flooding, heat stress or drought frequency and severity (Ipcc, 2019).



The influence of climate change on vegetation, especially on forest condition, is highlighted in several studies (Buras and Rammig and Zang, 2020; Schuldt et al., 2020; Prăvălie et al., 2022; Yang et al., 2019; Liu et al., 2015). It has the potential to

cause severe, long-term damage to forest ecosystems by increasing the frequency of extreme weather events, such as droughts, destructive windstorms, and wildfires in many regions. (Bryn and Potthoff, 2018; Hofgaard et al., 2012; Karlsen et al., 2017; Morin et al., 2018). That is why, monitoring vegetation dynamics and precisely characterizing the response of vegetation to changing climate is essential in order to maintain a sustainable environment (Tomlinson et al., 2011; Barbosa et al., 2019).

The most widely used parameter for evaluating vegetation's response to climate change is the normalized difference vegetation

index (NDVI), derived from satellite remote sensing (Adole and Dash and Atkinson, 2016; Huang et al., 2021; Soubry et al., 2021; Buras and Rammig and Zang, 2020; Barbosa et al., 2019). The NDVI is a normalized transform of the near-infrared to red reflectance ratio, which is intended to standardize vegetation index values to fall between −1 and +1 (Didan and Munoz, 2019). According to research, it is a trustworthy ecological indicator, if obtained from properly calibrated satellite-borne sensors (Huang et al., 2021). In the research of vegetation vigour, NDVI has a long history spanning 50 years, but in recent

times the enhanced vegetation index (EVI) has also gained popularity. In EVI formula the blue radiation is additionally used to stabilize the index value against variations in aerosol concentration levels (Didan and Munoz, 2019).

Spectral vegetation indices – NDVI and EVI – derived from Moderate Resolution Imaging Spectroradiometer (MODIS) data were coupled with meteorological elements in many research papers (e.g. Buras and Rammig and Zang, 2020; Li et al., 2010; Mao et al., 2012; Mbatha and Xulu, 2018; Moreira and Fontana and Kuplich, 2019; Zhu et al., 2023; Ghaderpour et al., 2023;

Schuldt et al., 2020). The applied coupling methods used in these studies were often based on single and multiple linear regressions and Pearson's correlations between vegetation indices and climate elements, but assuming the stationary relationship. However, the time lag in the correlation between vegetation indices and weather elements should not be disregarded. The spectral response to the influencing factor varies depending on the vegetation type – it is quicker for grasslands and agricultural lands (Moreira and Fontana and Kuplich, 2019), while in the case of forests this response might be

very extended in time (Barbosa et al., 2019; Carl et al., 2013), so a significant delay in correlation between vegetation condition and meteorological element can occur. For instance, elements such as temperature can influence the trees' phenological timing of the following year (Carl et al., 2013). Therefore, nowadays the wavelet coherence (WC) method is often used in order to

capture the delay in the spectral response of the vegetation. This method allows to study the multiscale and non-stationary

processes over finite spatial and temporal domains (Furon et al., 2008), and hence is advantageous when compared to the

Fourier transform, because the latter requires stationarity (Martínez and Gilabert, 2009). WC method has proven to be useful

in geophysics and climatology, linking e.g. rainfall and ENSO index (Torrence and Webster, 1999) or rainfall and monsoon

in Pakistan (Hussain et al., 2022). WC has already been used several times when coupling between climatological factors such

as temperature or rainfall and vegetation occurred. The coherence of meteorological elements and grasslands/savannas/forests

was researched e.g. in Brasil (Moreira and Fontana and Kuplich, 2019; Barbosa et al., 2019), South Africa (Mbatha and Xulu,

2018), southern China (Zhou et al., 2022), India (Naga Rajesh et al., 2023), or Indonesia (Erasmi et al., 2009). In Europe,

similar research was conducted in the Mediterranean (Ghaderpour et al., 2023). Surprisingly, the coherence between vegetation

dynamics and climate elements in the temperate zone is very understudied, and the existing studies are limited in time and

space (Carl et al., 2013; Zhu et al., 2022).

This study aims to identify patterns in time series of three different types of vegetation (broadleaved and coniferous forests

and pastures) in the temperate zone, and relate them with meteorological elements and teleconnection indices, using the

Wavelet Transform (WT) and Wavelet Coherence (WC). Thus, the main objectives of this research are: 1) to identify the

variability and periodic changes in time series of MODIS-based NDVI and EVI of different vegetation types, and in time series

of meteorological elements and teleconnection indices, using the WT method and 2) to couple the NDVI and EVI vegetation

indices with meteorological elements and teleconnection indices in order to determine the coherence and time lags in the

spectral response of individual vegetation types to the influencing factors, using the methods of WC and Pearson's correlation.

The analyses are carried out for the broadleaved and coniferous forests and pastures in the temperate zone of central Europe,

in the period 2002-2022.

## 2 Materials and methods

### 2.1 Study area

The study area characterised by three vegetation types – the two types of forest (broadleaved and coniferous forest) and pastures

including meadows and other permanent grasslands under agricultural use – is located in the administrative borders of Poland



(Fig. 1). The analysed vegetation types are situated within a territory extending from 49N to 54.5N latitude and from 14E to

24E longitude. From the north, the research area borders onto the Baltic Sea, while the terrain changes towards the south -

there are mountains at the southern edges of the research area. Because Europe's land relief is arranged mostly latitudinally,

there is no orographic barriers and climate in the study area is influenced by the western transfer of air masses, and therefore

indirectly by the Atlantic Ocean. The warm temperate climate is characterized by mean winter temperature from -3.5°C (in

the north-east and in the sub-mountain and foothill regions in the south) to 1.5°C (in the west), mean summer temperature

from 14.5°C (in foothill regions in the south) to 19.5°C (in the centre) and a mean annual precipitation sums from 450 mm in

the centre of the study area to 1200 mm in the mountains (1991–2020) (Tomczyk and Bednorz, 2022).

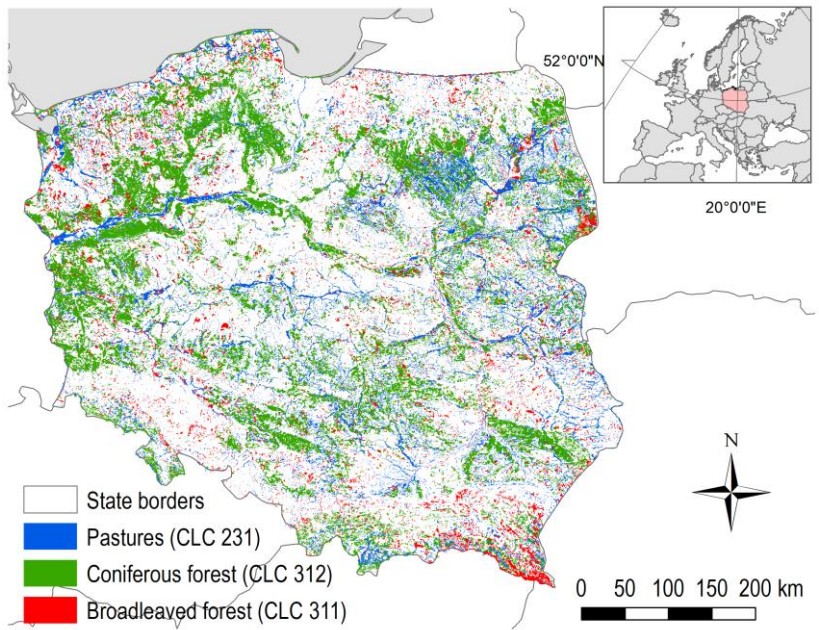

**Fig. 1. Spatial distribution of three vegetation masks – broadleaved forest (CLC class 311), coniferous forest (CLC class 312) and pastures (CLC class 231) – used in the study.**


The selected vegetation types were defined on the basis of Corine Land Cover (CLC) 2006, 2012 and 2018 databases (Clms,

2021). The CLC database provides data on land cover across 44 classes, in European countries. For areal phenomena, the CLC

employs a Minimum Mapping Unit (MMU) of 25 ha, and for linear phenomena, a minimum width of 100 m. (Clms, 2021).

The CLC 2006, 2012 and 2018 were used to prepare masks for broadleaved forests (CLC class 311), coniferous forests (CLC

class 312) and pastures (CLC class 231). The CLC forest vector layers for 2006, 2012 and 2018 were intersected and the

polygons that were still forest for these three periods made up the broadleaved or coniferous forest mask, respectively. The

percentage of forest coverage was calculated for each MODIS pixel. The pixels containing at least 80% of forest cover were

selected for further analysis. To ensure the uniformity of forest pixels, a criterion of 80% coverage of broadleaved or coniferous

forest was applied. Following these selection criteria, 174,243 pixels were retained as the broadleaved forest mask and 798,777

pixels were retained as the coniferous forest mask, representing the area of 10,890 km$^2$ and 49,924 km$^2$, respectively. Clusters

of broadleaved forest are rather small, and most of them are located in the north-western part of the study area, in the south-

eastern edge of the area (Bieszczady Mountains) and in the eastern part (Białowieża Forest). The tree species dominating in

the species composition are birch, oak and beech (2022). On the contrary, coniferous forest prevail in most of the study area,

and the predominant species, covering 58% of the forest area, is pine (2022). In the mountains, the proportion of spruce and

fir in stands species composition is also apparent (2022).

The pastures mask was prepared following the same steps, as used for forest masks, except that only CLC vector layers for

2012 and 2018 were used (because of the poor quality of the 2006 CLC class 231). Following such selection criteria, 338,193

pixels were retained as the pastures mask, representing the area of 21,137 km$^2$.

## 2.2 MODIS data – NDVI and EVI

This study uses two vegetation indices (VI) – the normalized difference vegetation index (NDVI) and enhanced vegetation

index (EVI) – derived from the Moderate Resolution Imaging Spectroradiometers (MODIS) onboard Terra and Aqua satellites

– products MOD13Q1 and MYD13Q1 (Didan, 2021b, a). Theoretical description of the MODIS VI and the NDVI and EVI

algorithm details are provided in Didan and Munoz (Didan and Munoz, 2019). The MOD13Q1 and MYD13Q1 products were

downloaded for the period 2002-2022. Because the data from Terra and Aqua is processed 8 days out of phase at 16-day

intervals, combining both satellites' data streams produces a quasi-8-day product time series (Didan and Munoz, 2019).

MOD13Q1 and MYD13Q1 products are published with 250 m spatial resolution. To cover the area between 49N and 55N

latitude and 14E and 24E longitude three granules were required, because each granule has 4,800 x 4,800 pixels. Eventually,

2,712 granules were needed to cover the time period 2002-2022.

Together with the NDVI (or EVI) product, the corresponding pixel reliability and day of year layers were used. Because in 16-

day composite the adjacent selected pixels may originate from different days, so for each pixel in such composite the day of

year layer keeps the information about the actual day the pixel originate, while the pixel reliability layer keeps the information

that describes overall pixel quality (Didan and Munoz, 2019). Based on this, only the pixels indicated as good or marginal

quality were selected, which is a common practice in similar studies (e.g. Buras and Rammig and Zang, 2020). In the next

step, based on the day of year information, each of the selected pixels was allocated to the respective month. To get the monthly

values of NDVI (or EVI), a monthly maximum NDVI (or EVI) was calculated for each of the retained pixels. The reason

behind this approach is that low-value observations are either erroneous or have reduced vegetation vigour for the time period

under consideration. (Holben, 1986).

Next, the deseasonalised time series of monthly anomalies from the multi-annual monthly values of NDVI (or EVI) were

prepared for each MODIS grid cell (i.e. each pixel), so that e.g. the deseasonalised value (anomaly) for January 2002 is the

difference between January 2002 value and multi-annual mean from all Januaries. It should be noted that the term "anomaly",

which is commonly used in climatological studies (e.g. Kulesza, 2021), should be interpreted as a "deviation from the mean

value". Finally, spatially averaged 252-element (21 years x 12 months) time series of NDVI (or EVI) anomalies in respective

vegetation masks were prepared. The spatially averaged values of NDVI (or EVI) were calculated as area averages of all NDVI

(or EVI) values in the MODIS grid cells (i.e. all pixels) within the respective vegetation masks. The methodology diagram

showing the above-described steps is presented in Fig. 2.

## 2.3 Meteorological elements

In this work, the gridded data from ERA5-Land reanalysis (Muñoz-Sabater, 2019, 2021) was used. The monthly data

representing meteorological elements, which are generally known to have a significant impact on the dynamics of vegetation

productivity (Chu et al., 2019; Liu et al., 2015; Yang et al., 2019), i.e. 2-metre temperature (T, in °C), precipitation (P, in mm)

and evapotranspiration (ETo, in mm) was downloaded for the period 2002-2022. Spatial extent of the meteorological data was

49N to 55N latitude and 14E to 24E longitude, and the resolution of reanalysis data was 0.1° x 0.1°. Additionally, monthly

data on 2-metre dewpoint temperature was downloaded in order to calculate the water vapour pressure deficit (VPD, in hPa),

a variable frequently used to explain the tree mortality (Gazol and Camarero, 2022; Schuldt et al., 2020). VPD is the difference



between saturation vapour pressure (SVP, which is temperature dependant) and actual vapour pressure (AVP, which is

dewpoint temperature dependant). SVP can be approximated from the air temperature records, following the Tetens' formula

(American Meteorological Society, 2023):

$$SVP = 6.11 \times 10^{\left(\frac{7.5T}{237.7+T}\right)}$$

and AVP can be calculated from the same equation, using dewpoint temperature instead of air temperature. Eventually, VPD

= SVP-AVP.

In the next step the deseasonalised time series of monthly anomalies from the multi-annual monthly mean values of T, P, VPD

and ETo were prepared for each grid cell of the ERA5-Land reanalysis. As in the case of NDVI (or EVI), the term "anomaly"

should be interpreted as a "deviation from the mean value". The data was then resampled to fit the MODIS grid cells (which

does not affect much the data accuracy, because monthly mean values of meteorological elements are slowly changing over

space). Finally, spatially averaged 252-element time series of T, P, VPD and ETo anomalies in respective vegetation masks

were prepared. The methodology diagram showing the above-described steps is presented in Fig. 2.

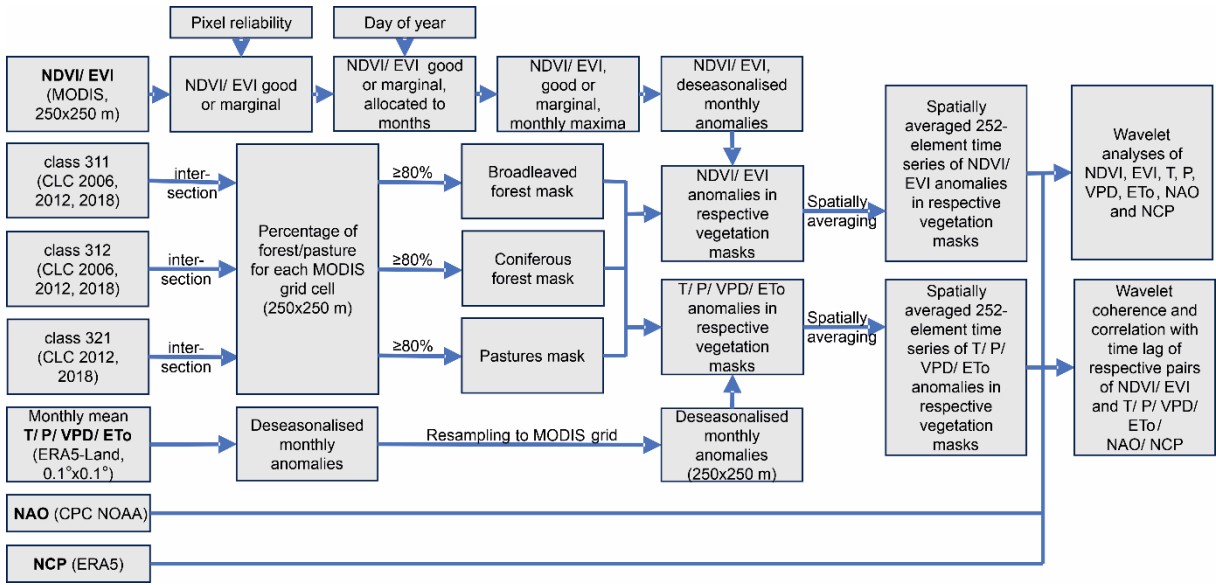

**Fig. 2. Flow chart of the input data and the methodology used in this paper.**



## 2.4 Teleconnection indices

This study uses two well-known teleconnection indices: North Atlantic Oscillation (NAO) and North Sea Caspian Pattern

(NCP). These large-scale climatic oscillations can be considered as a proxy of the general atmospheric circulation pattern,

giving aggregated information about the meteorological condition in a given year (e.g. drought-favouring conditions). The

relationship between teleconnection indices and vegetation condition in different regions of the world was the focus of several

studies (Brown and De Beurs and Vrieling, 2010; Gong and Shi, 2003; Vicente-Serrano and Heredia-Laclaustra, 2004; He et

al., 2022; Gouveia et al., 2008; Olafsson and Rousta, 2021). According to many research results, NAO is associated with NDVI

at higher latitudes in parts of the northern hemisphere (Vicente-Serrano and Heredia-Laclaustra, 2004; Olafsson and Rousta,

2021; Gouveia et al., 2008), while the NCP is associated with vegetation condition in western Eurasia (He et al., 2022).

Monthly values of NAO index in the period 2002-2022 were downloaded from Climate Prediction Center of the National

Oceanic and Atmospheric Administration ([https://www.cpc.ncep.noaa.gov/products/precip/CWlink/pna/nao.shtml](https://www.cpc.ncep.noaa.gov/products/precip/CWlink/pna/nao.shtml)). The

procedure used to calculate the NAO teleconnection index is based on the Rotated Principal Component Analysis (RPCA)

(Barnston and Livezey, 1987). The RPCA technique is applied to monthly mean standardized 500 geopotential height

anomalies in region 20N to 90N (and all longitudes) between January 1950 and December 2000. The anomalies are

standardized by the 1950-2000 climatology. In the positive phase of NAO the westerly circulation of the atmosphere prevails

over central and northern Europe, resulting in relatively warm and humid weather in winter, while cool and rainy in summer.

In the negative phase, the meridional circulation occurs more often, and central Europe can then be reached by cold and dry

air masses from the north or hot air masses from the south.

NCP index was calculated on the basis of 500 geopotential height monthly values derived from ERA5 reanalysis (Hersbach et

al., 2020), in the same period 2002-2022. The NCP index values were calculated from the normalised 500 geopotential height

difference between averages of North Sea (0E, 55N and 10E, 55N) and northern Caspian Sea (50E, 45N and 60E, 45N) regions

(Kutiel et al., 2002). In the negative phase, above normal temperatures and below normal precipitation occur in the Balkans,

western Turkey and the Middle East. In the positive phase – the other way round. There is no significant correlation between

the NCP and NAO (Araghi et al., 2019).





## 2.5 Methods

### 2.5.1 Wavelet analysis

The Wavelet Transform (WT) was applied to the deseasonalised time series of NDVI, EVI, T, P, VPD, ETo, as well as NAO

and NCP in searching for potential variations in frequency and time at different scales. To this end, the wavelet packet (Torrence and Compo, 1998) implemented to MATLAB computing environment was used. The use of wavelet analysis gives the information on fluctuations which change frequency over time. This is possible thanks to using wavelets – structures that are time-limited and consist of several short oscillations. The basic wavelet can be stretched and shifted in time, in order to create a so-called wavelet family – a collection of similar structures. Wavelet analysis is based on correlating the individual

elements of wavelet family with values of the time series throughout the observation period. The wavelet power spectrum – $|W|^2$ – represents this correlation. The higher the power, the wavelet will be more similar to the empirical data at a given point in the time series, which means that fluctuations of a given frequency are more likely to occur in a given period. In this paper, we used the Morlet wavelet and assessed the statistical significance of the $|W|^2$ values with the $\chi 2$ test (Torrence and Compo, 1998) (the level of significance $\alpha=0.05$). The regions of the wavelet power spectrum, which are especially vulnerable to adverse

edge effects (because of the finite length of the time series), are delimited by the 'cone of influence' (COI). The values of the wavelet power spectrum which are outside of COI, are considered uncertain.

### 2.5.2 Wavelet coherence and time lags

In order to determine the changing-over-time correlations between NDVI (or EVI) and meteorological elements and teleconnection indices, the wavelet coherence (WC) was applied to the deseasonalised time series of respective data sets,

resulting in 6 diagrams (scalograms) for each of the vegetation types for NDVI (for EVI likewise): NDVI with T, NDVI with P, NDVI with VPD, NDVI with ETo, NDVI with NAO and NDVI with NCP. Wavelet coherence combines the advantages of wavelet analysis and Pearson correlation, allowing for searching for correlations that vary over frequency and time (Torrence and Webster, 1999; Grinsted and Moore and Jevrejeva, 2004). In this paper, WC was prepared according to the Grinsted et al. (2004) in MATLAB computing environment. In the WC scalogram colour scale ranges from blue (low correlation) to red

(high correlation) and thus represents the wavelet coherence coefficient. The direction of arrows indicates the phase delay between signals (time series): right arrows indicate that the series are completely in phase, i.e. positive correlations, while the



left arrows indicate that the series are completely out of phase, i.e. negative correlations. Statistical significance of values of the wavelet coherence coefficient was assessed using Monte Carlo method, at the significance level of α=0.05 (Grinsted and Moore and Jevrejeva, 2004).

In order to additionally investigate the delays in the spectral response of the individual vegetation type to the triggering meteorological factors, the overall, linear correlations with appropriate time lags were calculated. The correlated pairs of data sets were prepared with 0 to 36 months delay (3 years). 0-month delay means that independent variable's values (T, P, VPD, ETo, NAO, NCP) from month $i$ were correlated with dependent variable's values (NDVI, EVI) from the same month. In turn, 1-month delay means that independent variable's values from month $i$ were correlated with dependent variable's values from

the $i$+1 month, and so on. The strength of the correlation between the deseasonalised time series of NDVI (or EVI) and meteorological elements and teleconnection indices for three vegetation types was assessed using the Pearson correlation coefficient, expressed by the following formula: $r = {cov_{xy}}/{S_x S_y}$, where $cov_{xy}$ is the covariance in the bivariate distribution of the variables $x$ (time series of a respective meteorological element or teleconnection index) and $y$ (time series of NDVI or EVI), $S_x$ and $S_y$ are the standard deviations in the marginal distributions of the variables $x$ and $y$ respectively. The significance

of linear correlations calculated in this way was assessed at the significance levels of α=0.05.

### 3 Results

#### 3.1 Basic characteristics of VI and meteorological elements

In the last two decades a slightly positive trend in condition of the forests in Poland was noticed, with a mean NDVI increase of 0.088×10[-3] per year (2002-2021) (Kulesza and Hościło, 2023). The biggest increase of mean annual NDVI (by 0.030 in 20

years) was observed in central-eastern Poland, while it was weaker in southern, western and northern edges of the study area. In turn, the biggest mean annual NDVI was observed in forests of the foothill regions in the south and also in the Baltic Sea coastal region, while central regions had lower NDVI. In general, broadleaved forests had slightly bigger mean NDVI (0.841) than coniferous forests (0.791).

The trend of mean annual T was positive over the entire research area, resulting in the increase in T by 1 to 1.6°C (in southern

and eastern regions) (2002–2021) (Kulesza and Hościło, 2023). In central and eastern regions the statistically significant

increase in ETo was also reported, with a mean increase of 1.79 mm per year (while mean annual ETo is ca. 600 mm). The

slope of the trend in changes in P appeared insignificant in the whole study area. For the detailed analysis of the spatiotemporal

variability and trends in NDVI and T, P, ETo over Poland in the last two decades the reader is referred our previous paper

(Kulesza and Hościło, 2023).

The course of the monthly anomalies of NDVI during the period 2002-2022 showed the dynamics of three vegetation types.

Positive anomalies of NDVI in the growing season (April-September) were noticeable in 2011, 2013, 2016 and 2021, whereas

the negative anomalies of NDVI occurred in the growing season of 2003, 2008, 2018, 2019 and 2022 (Fig. 3). In 2015, the

negative peak of NAO index in July caused the positive T anomaly, negative P anomaly and very big, negative VPD anomaly

(i.e. bigger-than-average deficit of water vapour) in August. In turn, all this resulted in negative values of pastures' NDVI in

August and September of 2015, but forest condition seemed unaffected. In 2018, the combined effect of above-average T (in

warm half-year) and mostly below-average P resulted in gradually decreasing values of NDVI in the growing season. Yet, the

decrease in NDVI values was not big. Additionally, generally positive T anomaly and negative P anomaly in the growing

season of 2018 resulted in big, negative VPD anomaly (i.e. deficit of water vapour bigger than average) and big, positive

anomaly of ETo. The following year (2019), experienced similar meteorological condition (although not so severe), but the

vegetation condition during the growing season was significantly below average. Unlike in 2018, when NAO phase was

positive in the growing season, in 2019 the NAO phase was negative during the whole growing season. Moving forward, year

2021, probably because of the above-average P in April, May and August, experienced the positive anomalies of vegetation

condition for all three types of vegetation. On the contrary, the following year (2022) experienced the negative anomalies of

NDVI values, especially visible at the beginning of the growing season (April-May), and especially severe for pastures. In

March, May and June of 2022, there were significant negative anomalies of P and VPD, together with positive anomalies of

ETo.





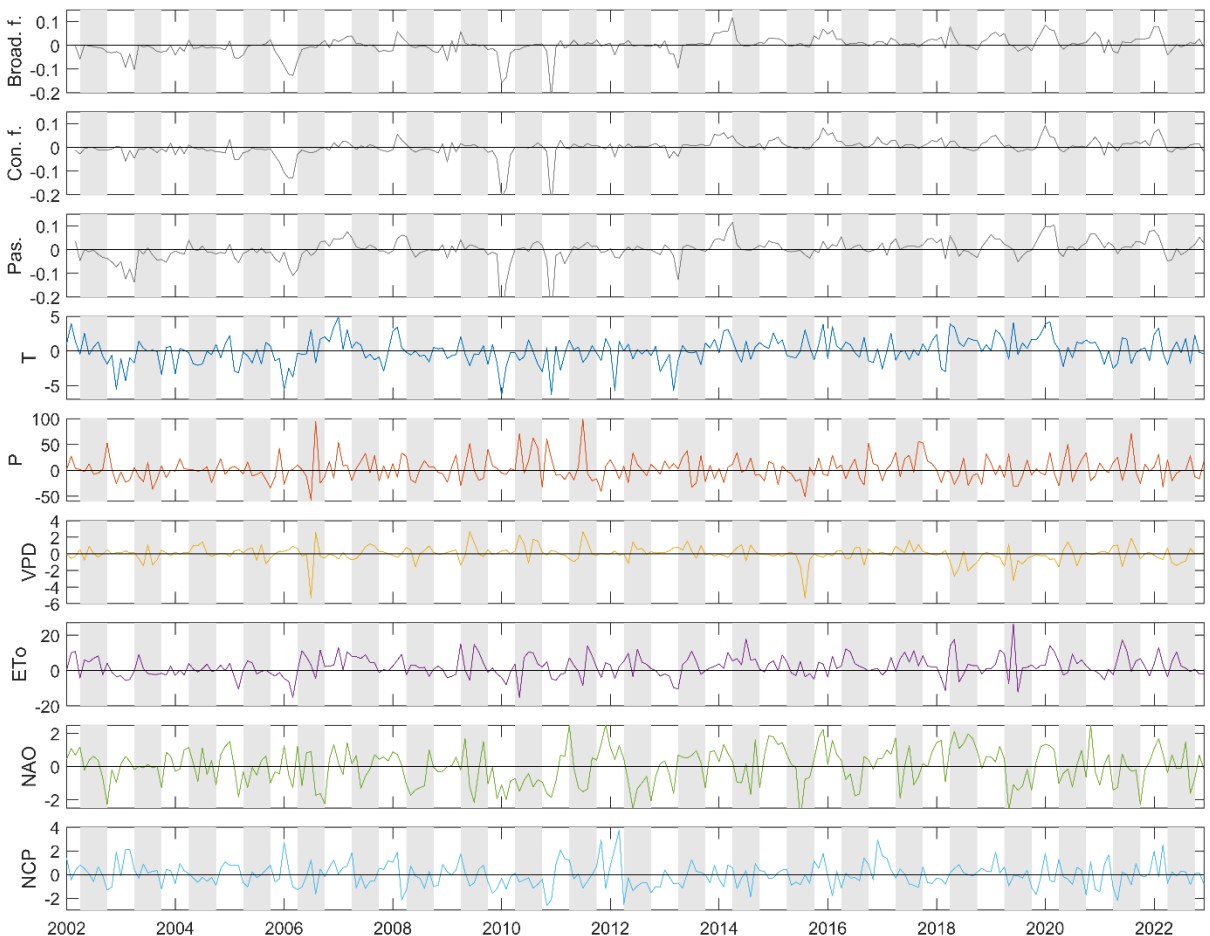

**Fig. 3. The deseasonalised time series of monthly anomalies of NDVI (broadleaved forest, coniferous forest, pastures), T, P, VPD, ETo, and NAO and NCP indices, in the period 2002-2022. Grey areas refer to warm half-years (April-September).**

**3.2 Variability and periodic changes in VI and meteorological elements and teleconnection indices**

The data sets used in the study were purposely deseasonalised, so the obvious 1-year cycle in both NDVI (or EVI) and meteorological conditions is removed. Thus, WT was used in searching for cycles and fluctuations with lower or higher frequency over time (i.e. interannual or intraannual cycles). Consequently, no strong cycles are visible in the graphs which show the wavelet power spectrum $|W|^2$ (Fig. 4 and 5). The pulse of a half-year and 1 year cycle of fluctuations in NDVI is marked around the 2010 for all three types of vegetation (Fig. 4, left column). Although they are statistically significant, neither



the power spectrum is strong, nor they last long. The EVI shows similar pattern for pastures, but much fewer statistically

significant fluctuations for broadleaved and coniferous forests (Fig. 4, right column).

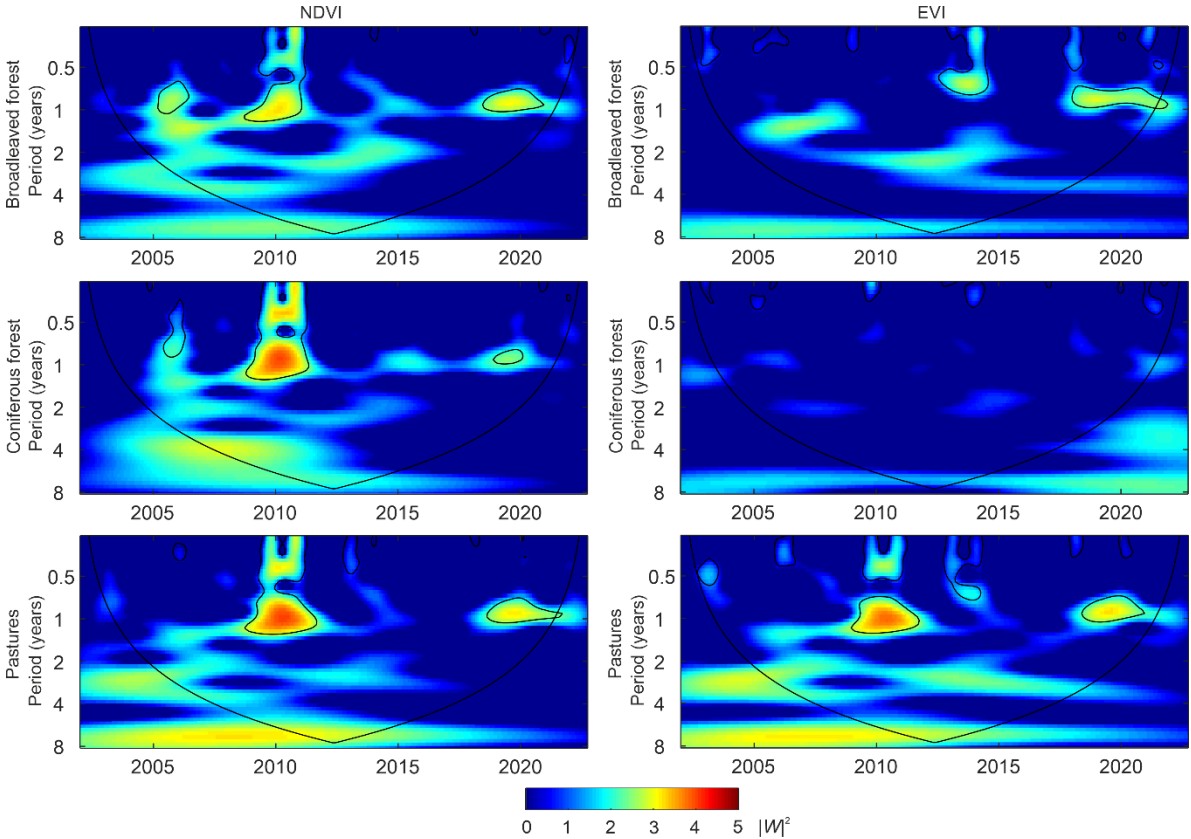

**Fig. 4. Wavelet power spectrum ($|W|^2$) for deseasonalised time series of NDVI (left column) and EVI (right column) for different vegetation types during the period 2002-2022. The COI region is below the thick black line. Statistically significant areas at the level of α = 0.05 are indicated by a thin black line.**

Meteorological elements also do not show significant interannual cycles. The components with a period of less than half-year

are more visible, but, similarly as in the case of NDVI, although they are statistically significant, the power spectrum is rather

weak (Fig. 5). Only VPD shows a cyclical component of circa 4 years, but it lies partly in the COI region and is statistically

insignificant. NAO and NCP produce significant components with a period of less than half-year (weak power spectrum), and

additionally a short pulse of a 1 year cycle that is visible around 2011 (NAO and NCP) and 2015 (NAO only) (Fig. 5, lower

panel).




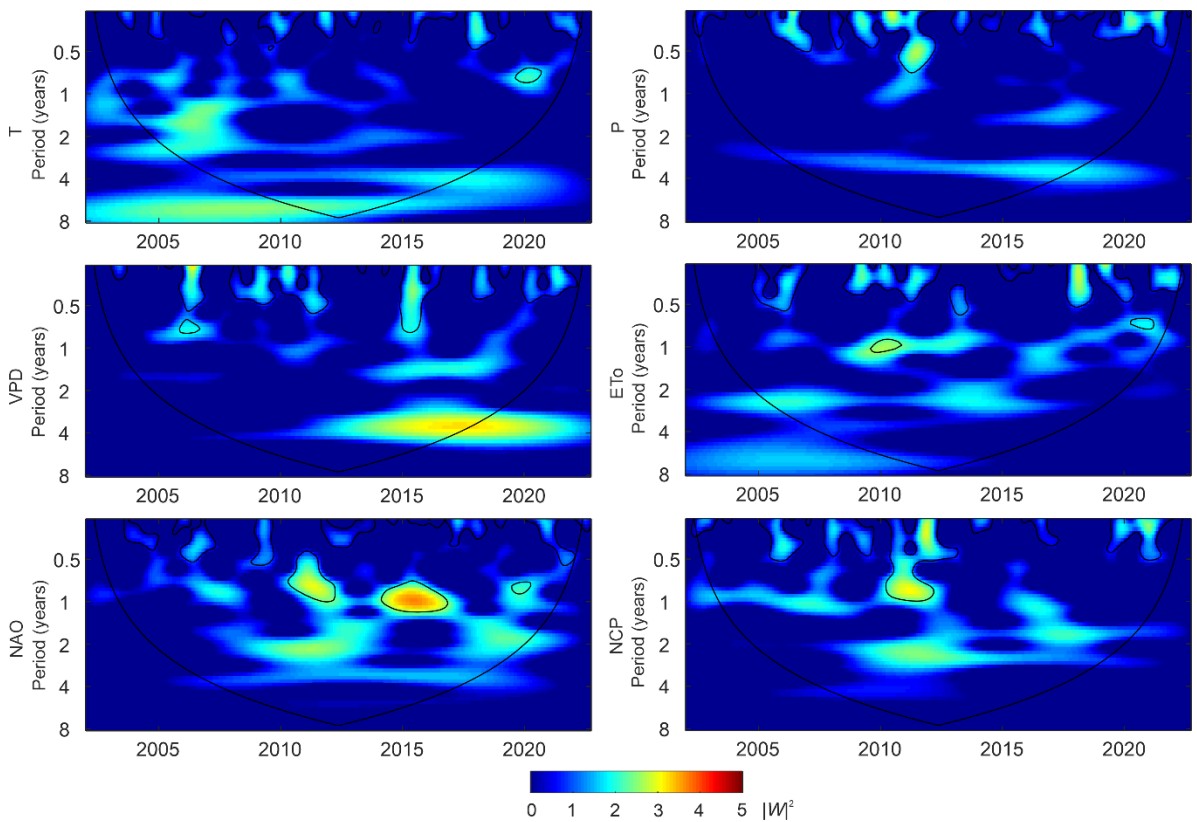

**Fig. 5. Wavelet power spectrum ($|W|^2$) for deseasonalised time series of T, P, VPD, ETo (mean value from three vegetation masks), and NAO and NCP during the period 2002-2022. The COI region is below the thick black line. Statistically significant areas at the level of $\alpha = 0.05$ are indicated by a thin black line.**

### 3.3 Coherence and time lags in the spectral response of individual vegetation types to the influencing factors

The pattern observed for coherence between NDVI and meteorological elements and teleconnection indices is different for each of these factors.

T shows high coherence with the NDVI in all three types of vegetation. There is a significant common power in the 8-16 month (circa 1 year) band for the periods 2010-2015 and 2020-2022 (Fig. 6). Second significant common power band is 20-32 months (circa 2 years), visible for both forest types for the whole time period 2002-2022 (and for pastures only up to 2016). The high cohesion values indicate that the data sets exhibit high correlation in a year given in the *x* axis, and with a delay given in the *y*



axis. The observed regularities are additionally proven by the Pearson's linear correlations with appropriate time lags.

Significant positive correlations between NDVI and T occur for broadleaved and coniferous forests for 8-, 12-, 27-, 28-, 29- and 30-month delay (Fig. 8). For pastures significant positive correlations between NDVI and T only occur for 12- and 27- month delay.

Both P and VPD produce rather weak coherence with NDVI in all three vegetation types. Very small patches of high coherence of circa 1 year delay between NDVI and P occur only around 2006 and 2009-2010 for broadleaved and coniferous forests (and

for pastures only one patch around 2009-2010) (Fig. 6). VPD produces even smaller patches of high coherence of circa 8- month delay (around 2006), in where the NDVI and VPD are mostly out of phase, meaning that the correlation between them is negative (Fig. 6). In fact, significant negative Pearson's correlations appear for 7-, 8- and 9-month delay for all vegetation types (Fig. 8). Fig. 8 indicates also significant correlations between NDVI and VPD for 18- and 22-month delay.

ETo shows high coherence with NDVI in all three vegetation types. The significant common power appears in the intraannual

(3-8 months) band, from the beginning of the study period until 2008 (Fig. 6). High and significant coherence of circa 1 year (8-16 month) delay occurs mostly around 2010, while significant coherence of circa 2 year (20-32 month) delay is distributed more or less along the whole study period. Surprisingly, it seems the most stable for pastures, which is low grassy vegetation, rather independent from interannual weather conditions. Significant positive Pearson's correlations between NDVI and ETo occur for broadleaved and coniferous forests for 8- and 22-month delay, while for pastures only for 22-month delay (Fig. 8).

Correlations in specific bands – intraannual and interannual – are also visible between NDVI and NAO index in particular time periods. NAO produces strong coherence with NDVI mostly for two forest types. Small areas of high positive correlation of circa 1 year delay between NDVI and NAO appear mostly for coniferous forest in the period 2013-2016 and 2018-2021, as well as for broadleaved forest in the period 2015-2016 and 2019-2020. This is additionally proven by the significant positive Pearson's correlation between NDVI and NAO for 11-month delay (Fig. 8). For pastures this interannual pattern is hardly

recognizable (Fig. 6). For broadleaved and coniferous forests the coherence of circa 3 year delay up till 2013 is also visible. Interestingly, there is a significant common power for NDVI and NAO in the 2-6 month (intraannual) band for the year 2018 in all three vegetation types. Similar, small patches of high and significant, intraannual coherence for this year are mostly



visible for broadleaved forest regarding T, VPD and ETo, while for coniferous forest and pastures regarding ETo only (Fig. 6).

NCP index produces rather weak coherence with NDVI in all three vegetation types. The cohesion pattern is somehow similar to the one produced by NAO, with small areas of high and significant coherence of circa 1 year delay around 2015 and 2020, which is mostly visible for coniferous and broadleaved forests (Fig. 6). Additionally, for two forest types the coherence of circa 32 month (almost 3 year) delay in the period 2010-2015 is also visible. Surprisingly, NCP's Pearson's correlation with NDVI is significant for 25- and 35-month delay (for forest types), but, unlike the cohesion's right arrows, this correlation is

negative (Fig. 8).

The pattern observed for coherence between EVI and meteorological elements and teleconnection indices resembles in many places the pattern observed for NDVI, especially regarding pastures.

However, in forest types, concerning T and ETo, their coherence with EVI gives substantially smaller areas of high and significant cohesion, as compared to NDVI (Fig. 8). Nevertheless, the Pearson's linear correlations with time lags, prepared

for EVI, reveals many significant and positive correlations between EVI and T for broadleaved forest (mostly 1 year delay and 2 year delay), and even more significant correlations for coniferous forest. In turn, significant positive Pearson's correlations between EVI and ETo occur in broadleaved forest for circa 2 year (22-month) delay, while in coniferous forest for circa 1 year (10-month) delay and 2 year (22- and 23-month) delay (Fig. 8).

P and VPD show some more areas of significant common power with EVI than with NDVI, but the high cohesion areas are of

over 3 year delay and lie partly in the COI region (Fig. 8).

In case of NAO, there is a significant common power with EVI in the 30-40 month (circa 3 year) band in the period 2005-2020, for broadleaved forest (Fig. 7). This is additionally proven by the Pearson's linear correlations with 30-month time lag (Fig. 8). Similarly as in the case of NDVI, there is a significant common power for EVI and NAO in the 2-4 month (intraannual) band for the year 2018 for both forest types. Similar, small patches of high and significant, intraannual coherence for this year

are visible for broadleaved and coniferous forest regarding T, VPD and ETo (Fig. 7).

NCP shows less areas of significant common power with EVI than with NDVI. However, there are some significant positive Pearson's correlations between EVI and NCP, but there are also significant negative correlations (30-month delay for



broadleaved forest) (Fig. 8). Moreover, unlike other meteorological variables, here the arrows on the WC scalograms tend to

orientate up or down, which could be interpreted as uncoupling between both signals (Fig. 7).




**Fig. 6. Wavelet Coherence power spectrum (colour scale) between deseasonalised time series of NDVI and T, P, VPD, ETo, NAO and NCP for broadleaved forest (left column), coniferous forest (middle column) and pastures (right column) during the period 2002-2022. Colours range from blue (low correlation) to yellow (high correlation). Arrows indicates the phase difference between signals: right arrows – series are completely in phase, left arrows – series are out of phase. The COI region is below the thick black line. Statistically significant areas at the level of α = 0.05 are indicated by a thin black line.**





**Fig. 7. Wavelet Coherence power spectrum (colour scale) between deseasonalised time series of EVI and T, P, VPD, ETo, NAO and NCP for broadleaved forest (left column), coniferous forest (middle column) and pastures (right column) during the period 2002-2022. Colours range from blue (low correlation) to yellow (high correlation). Arrows indicates the phase difference between signals: right arrows – series are completely in phase, left arrows – series are out of phase. The COI region is below the thick black line. Statistically significant areas at the level of α = 0.05 are indicated by a thin black line.**




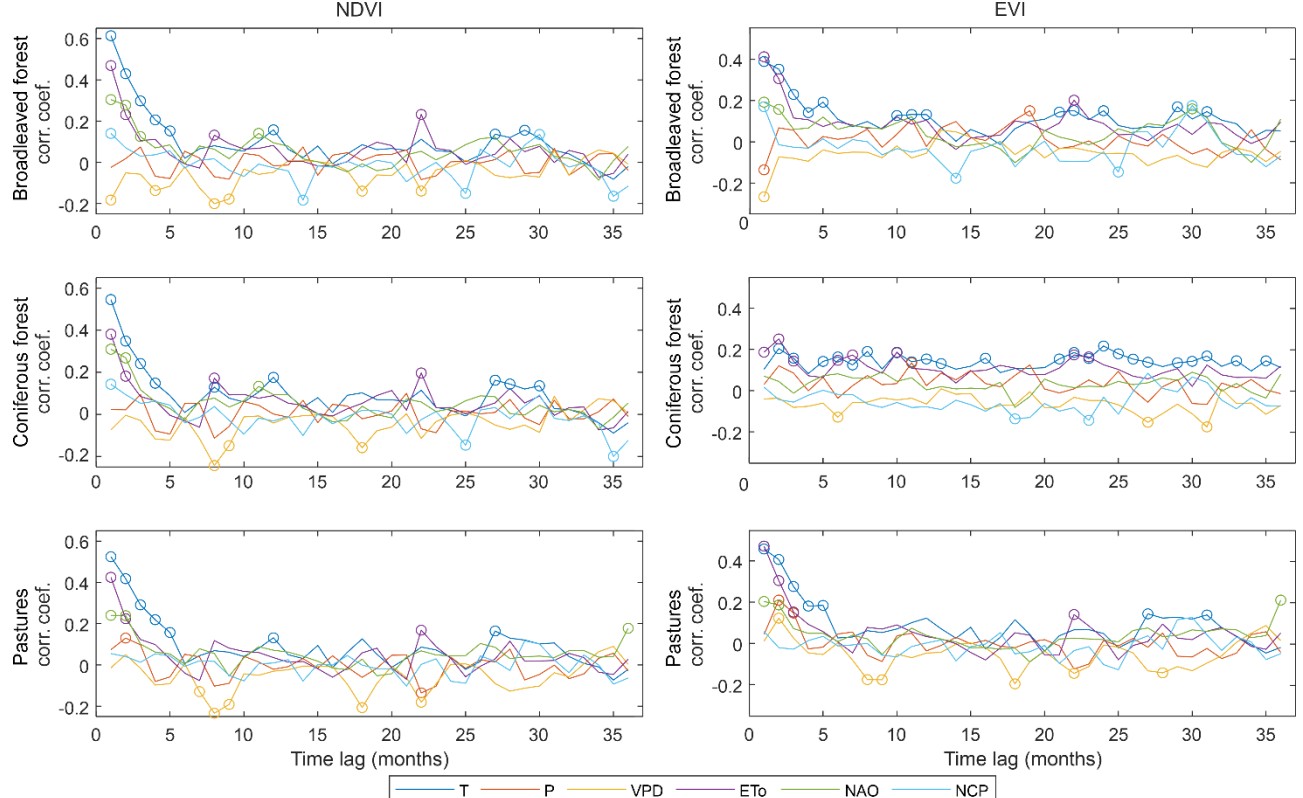

**Fig. 8. Correlation coefficients between deseasonalised time series of NDVI (left column) or EVI (right column) and T, P, VPD, ETo, NAO and NCP for three different vegetation types: broadleaved forest (upper panel), coniferous forest (middle panel) and pastures (bottom panel) The pairs of data sets are correlated with appropriate time lag of 0 to 36 months. Statistically significant correlations at the level of α = 0.05 are indicated by a circle.**

## 4 Discussion

Observed years with decreased vegetation condition, and unfavorable meteorological conditions related to this (above-average T and below-average P) are identified in many research studies. A number of severe, large-scale drought events occurred across Europe in the last 20 years, many of which have also affected the area of this study. In 2003, a severe drought mostly affected south-western Germany, Switzerland and south-eastern France (Fink et al., 2004; García-Herrera et al., 2010) with less impact on Poland (Somorowska, 2022). As the NAO index was oscillating around zero, the anticyclonic pattern that led

to a drought corresponded more to anomalous northern displacement of the Azores High than a typical blocking structure (Fink et al., 2004). Yet, the key factor to reach unprecedented temperature anomalies was soil moisture deficit (Fink et al., 2004).



Indeed, peak of positive ETo was also visible in May 2003 in our study area. In turn, in 2015 the negative peak of NAO index in July caused the drought in Europe that reached its peak intensity and spatial extent in August, affecting especially the eastern part of Europe (Ionita et al., 2017). Here, a very big, negative VPD anomaly occurred in August and resulted in decreased

condition of the vegetation, especially of pastures. Another severe droughts occurred in Europe in 2018 (Buras and Rammig and Zang, 2020; Schuldt et al., 2020; Boergens et al., 2020), 2019 (Boergens et al., 2020; Hari et al., 2020) and 2022 (Buras and Meyer and Rammig, 2023; Wang et al., 2023). All of them were observed also in our study area. In summer 2018 daily maximum temperature in Poland was 3.3°C higher than the 1981-2019 average (and 1.2°C higher than daily maximum temperature in 2003), and precipitation was below average as well (Somorowska, 2022). In 2019, the negative phase of NAO,

persisting for whole growing season of 2019, contributed to anticyclonic circulation and southerly advection of the air masses over Poland. The frequency of circulation from the south and south-east direction was 2 to 2.5 times higher than the average for the 1951-2018 (Ziernicka-Wojtaszek, 2021). As a consequence, June 2019 with a positive temperature anomaly of 5.0°C for the whole Poland was the warmest month since 1951 (Ziernicka-Wojtaszek, 2021). The occurrence of the consecutive summer droughts in 2018 and 2019 was unprecedented, and its combined impact on the growing season vegetation was much

stronger compared to the year 2003 (Hari et al., 2020). Indeed, the negative anomalies of NDVI were observed for all three vegetation types in 2018, but the vegetation condition during the growing season of 2019 was significantly below average. It is also important to mention that increased vegetation growth at the beginning of the growing season, caused by favorable meteorological conditions, contributes to fast depletion of resources (e.g. soil moisture) and promotes and strengthens droughts in summer (Somorowska, 2022; Bastos et al., 2020). That was the case of 2018 and 2019 too.

On the other hand, a strong heatwave and resulting flash drought from 2010 (Christian et al., 2020), as well as western European drought from 2017 (García-Herrera et al., 2019) were not noticeable in Poland (Somorowska, 2022). In 2010, though the NAO index was negative for the whole growing season, the NDVI anomalies in the study area were very small and positive for late summer and autumn, thanks to the relatively big sums of P. The vegetation condition in 2017 was even better, with positive NDVI anomalies from April until October.

In order to conduct the WT analysis, both NDVI (or EVI) and meteorological time series were purposely deseasonalised, so the obvious 1-year cycle, resulting from the seasonality of weather pattern and vegetation in temperate zone, is removed. Time

series of meteorological elements do not show any significant interannual cycles in the period 2002-2022. Similarly, no significant interannual cycles in meteorological time series in central Europe were found in other works, using much longer time periods, regarding T (Sen and Ogrin, 2016) and P (Brázdil et al., 2021; Sen and Kern, 2016). On the contrary, a short pulse of a 1 year cycle in the time series of NAO, occurring around 2011, was observed by e.g. Schulte et al. (Schulte and Duffy and Najjar, 2015). However, most features in its wavelet power are indistinguishable from a red-noise background, suggesting that NAO is rather a stochastic, unpredictable and a strong non-stationary process (Schulte and Duffy and Najjar, 2015; Pozo-Vázquez et al., 2001). Also, no strong, interannual cycles are visible in the deseasonalised time series of NDVI (and EVI). Overall, thanks to conducting the WT analysis, no significant intra- or interannual cycles were detected in both vegetation (NDVI and EVI) and meteorological variables. Thus, presumably the spectral response of the vegetation to the triggering meteorological factors is caused rather by the actual relationship between these variables, than is the result of a coincidence. The WC and Pearson's linear correlation with appropriate time lags helped revealing these relationships between NDVI (or EVI) and meteorological elements.

According to many scientific studies, temperature is one of the most influential elements, shaping the vegetation condition worldwide (Moreira and Fontana and Kuplich, 2019; Ghaderpour et al., 2023; Mbatha and Xulu, 2018). The correlation between grasslands' vigour and air temperature was strong in the annual cycle in southern Brasil (2000-2014) (Moreira and Fontana and Kuplich, 2019). Similar, significant common power in 8-16 month (ca. 1 year) band was produced by NDVI and soil temperature for savannas and forest in South Africa in the period 2002-2017 (Mbatha and Xulu, 2018). In Europe, significant annual in-phase coherency was observed between NDVI and land surface temperature in northern Italy in the period 2000-2021 (Ghaderpour et al., 2023). In temperate zone analysed in this study, T shows high coherence with NDVI in both forests and pastures. There is significant cohesion with 8-16 month (ca. 1 year) delay and 20-32 month (ca. 2 year) delay. Pearson's linear correlation shows more time lagged significant correlations between NDVI (or EVI) and T for forests than for pastures. This suggests that there is a significant lag in the forests' response to the changes in T, more noticeable for forests than for pastures. It is in line with the results of other researchers. For instance, in a beech forest in Germany, a time shift of approximately 300 days appears in the WC between NDVI and T (1989-2007) (Carl et al., 2013).



Similar, high coherence values are produced by ETo in both forest types and in pastures – mostly coherence of circa 1 year (8-16 month) and circa 2 year (20-32 month) delay. Indeed, forest vegetation response to water deficit can be delayed, especially depending on the tree type (broadleaved or coniferous) and species. Similar, in-phase relationship in the 8-18 month band occurred in savannas and forest in South Africa (Mbatha and Xulu, 2018). Yet, the surprisingly high coherence of 2 year delay,

which occurs for pastures along the whole study period is interesting, because low grassy vegetation seems to be unrelated to interannual weather conditions. However, one should remember that in statistics, a significant correlation between two variables may occur by chance, so a significant commonality in a wavelet coherence spectra analysis does not necessarily imply interconnection (Mbatha and Xulu, 2018).

At this point, it is also worth adding some comment on the differences in WC pattern observed for NDVI and EVI. Regarding

pastures, the coherence between EVI and meteorological elements and teleconnection indices resembles (more or less) the pattern observed for NDVI, while it differs substantially when forest types are concerned. Indeed, the correlation coefficient between deseasonalised anomalies of NDVI and EVI is very high for pastures (0.94), while lower for broadleaved forest (0.67) and coniferous forest (0.28). It is also worth noticing that regarding forest, the EVI values are within much narrower range than values of NDVI (Fig. 9), mostly because of the very low NDVI values in winter months of 2006 and 2010. The reason

for this might lie in different parameters used to construct both indices. The research results indicate that EVI is more susceptible to changes in canopy structure, while NDVI is more sensitive to chlorophyll (Huete et al., 2002). Thus, while EVI may more accurately represent early leaf shedding, NDVI is more likely to represent changes in leaf colour, such as those occurring during premature leaf senescence during a drought (Buras and Rammig and Zang, 2020). While some researchers suggest that NDVI reflects natural vegetation better than EVI (Li et al., 2010), others prefer the latter, which also uses blue

radiation to stabilise the index value against variations in aerosol concentration levels (Didan and Munoz, 2019).





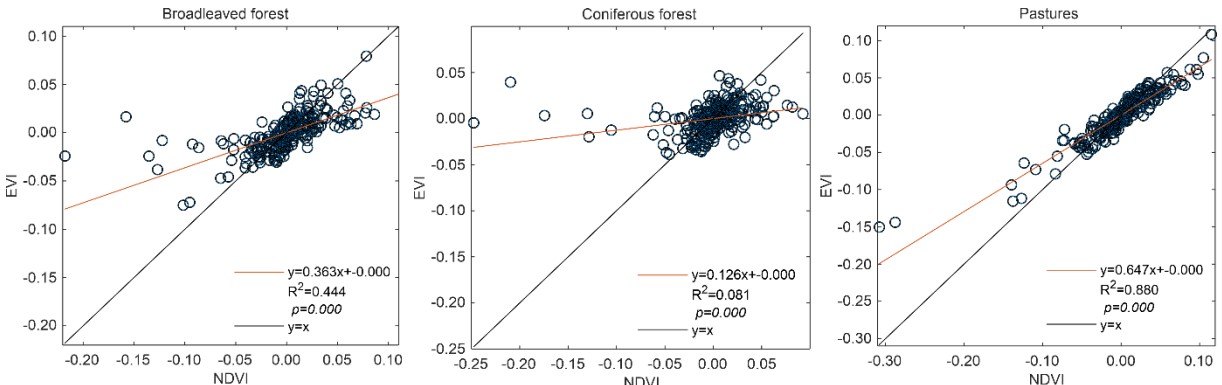

**Fig. 9. Scatterplots of deseasonalised anomalies of NDVI and EVI for three different vegetation types: broadleaved forest, coniferous forest and pastures. The graphs also show the linear regressions (red lines), the regression equations, the coefficient of determination $R^2$, and the statistical significance level $p$.**

Unlike in tropical and subtropical zones, P seems to have weaker coherence with NDVI in European temperate zone. Lotsch et al. (2003) showed that ecosystems in arid and semi-arid climate regimes (shrublands, savannas, grasslands) are most

sensitive to seasonal P anomalies at time scales of 4-6 months, whereas forested land areas exhibit weak correlation with P anomalies at all time scales. According to Lotsch et al. (2003) European temperate zone lies in the area of weak correlations between P and NDVI. The relatively short time lag in vegetation response to P anomalies was reported e.g. in Brazil: Barbosa et al. (2019) observed that *Caatinga* vegetation responds to P with a 1-3 month lag, which varies depending on the vegetation type and terrain characteristic (2008-2016). In southern Brazil, low coherence between EVI and P was observed only in regions

where dry periods may occur during summer (thus similar to temperate zone weather conditions) (2000-2014) (Moreira and Fontana and Kuplich, 2019). In Italy in 2000-2021, NDVI and P were significantly coherent only in the south, where the climate is the driest (Ghaderpour et al., 2023). In general, vegetation in subtropical or tropical zone is more weather-dependent, leading to high coherence between vegetation vigour and P anomalies. On the other hand, the temperate forest is less dependent on water availability, thanks to the deep root system that trees can use to reach deep water resources. However, it makes the

influence of meteorological factors on forest NDVI/EVI less evident and harder to detect.

Another important characteristic of the vegetation in temperate zone is that in winter vegetation mostly pauses (apart from some coniferous tree species), while meteorological elements obviously do not. P shows not only a seasonal variability (bigger sums of P occur in summer and lower in winter), but its intra- and interannual variations in winter are relatively high. Resulting

coherence between NDVI and P on the monthly basis, detected on the WC scalogram is rather weak. To compare, the NDVI

correlates well with P on the yearly basis, and this correlation is much stronger than the one between NDVI and T (Kulesza

and Hościło, 2023).

Drought stress on vegetation over Europe is linked also to various teleconnection patterns, with most studies focusing on the

influence of NAO on vegetation condition (Gouveia et al., 2008; Olafsson and Rousta, 2021; Araghi et al., 2019). According

to Gouveia et al. (2008), negative values of winter NAO induce low values of NDVI in spring, but high values of NDVI in

summer in northeastern Europe. Positive phase of NAO has the opposite effect, respectively. This behaviour mainly results

from the strong impact of NAO on winter temperature, associated with the critical dependence of vegetation growth on the

combined effect of warm conditions and water availability during the winter season (Gouveia et al., 2008). In this study, it was

especially visible in the year 2018, when the massive drought over Europe occurred. The NAO-induced stable high pressure

system formed over central Europe in April, and lasted until October 2018, causing exceptionally high temperature and big

deficit of water vapour. The spectral response of the vegetation to changes of NAO index was 2-6 months delayed – and similar

were the delays in response to changes of T, VPD and ETo.

However, apart from this exceptional drought of 2018, when the vegetation response to the triggering factors was relatively

quick, NAO produces strong coherence with NDVI mostly for forests, with circa 1 year delay and – a weaker coherence – with

circa 3 year delay. For pastures these interannual patterns are hardly recognizable.

On the contrary, NCP index produces rather weak coherence with both NDVI and EVI in all three vegetation types. As there

are both positive and negative (depending on the time lag) significant Pearson's correlations between NDVI (or EVI) and NCP,

it seems that the overall influence of NCP on vegetation condition in European temperate zone is rather weak. It is contrary to

the results of He et al. (2022), obtained for western Eurasia for 1981-2015. They concluded that NCP has significant negative

impact on meteorological and vegetation conditions over this region. According to He et al. (2022) the positive NCP phases

may better contribute to drier conditions over the region than NAO, because the positive phases of NCP contribute to increasing

dryness, thus causing the region to become more water-limited. In other parts of the world NCP's influence on vegetation is

also reported. It is the most influential teleconnection index affecting phenological metrics of forest and grassland vegetation

in Iran, both without a lag and with a lag of 1 year (1982-2015) (Araghi et al., 2019).

Finally, it should be noted that the conducted analyses have a few limitations. In general, vegetation in warm temperate climate

is highly seasonal. In the face of that, a severe weather condition occurring at the beginning of the growing season (e.g. drought), can induce poor vegetation condition in summer and autumn. However, big positive anomalies of T, or big, negative anomalies of P that occur in late autumn, have much smaller influence on vegetation condition in winter. That is why the intraannual relationships, with the time lag smaller than 1 year are much harder to detect than similar relationships in the tropical or subtropical zones.

Another important issue is the interpretation of the spectral indices' values. For instance, the NDVI signal coming from forest reflects not only the trees vigour, shaped by weather conditions, but also the "noise" from the understorey and other effects like pests, herbivores, pathogens and forest management. Grassland seem to be mostly free from such problems. Its response to the triggering meteorological factor is usually quick. Also, the quality of grassy vegetation in one year does not have a major effect on its quality in the subsequent year. In this study we only used pastures (CLC class 231) and not arable lands (CLC

class 211 and 212) or natural grasslands (CLC class 321) (Clms, 2021) to ensure the uniformity of the grassy vegetation. Natural grassland class in Poland is assigned mostly to small areas of military training grounds and alpine grassland with rough, uneven ground, steep slopes and up to 50% of bare rocks or bare natural surfaces (Kosztra et al., 2017). However, pastures are mowed during the growing season, which changes their spectral properties regardless of the weather. Because of this, proper interpretation of the obtained results can be difficult. At the same time, such results should be treated with caution.

**5 Conclusion**

The results presented in this paper show in detail the coherence and time lags in the spectral response of three individual vegetation types in temperate zone to the influencing meteorological factors, in the period 2002-2022. Vegetation condition in broadleaved forest, coniferous forest and pastures was measured with monthly anomalies of two spectral indices – NDVI and EVI. As meteorological elements we used monthly anomalies of T, P, VPD, ETo, and teleconnection indices NAO and NCP.

Periodicity in the time series of different vegetation types, and in the time series of meteorological elements and teleconnection indices, was assessed using the WT method. In turn, coherence between NDVI/EVI and meteorological elements was described

using the methods of WC and Pearson's linear correlation with time lag. The use of various research methods helps to objectify the results obtained.

Thanks to conducting the WT analysis, no significant intra- or interannual cycles were detected in both vegetation (NDVI and EVI) and meteorological variables.

In European temperate zone analysed in this study, the weakest coherence with vegetation condition is produced by P and VPD. Also NCP produces rather weak coherence with both NDVI and EVI in all three vegetation types. On the contrary, NAO produces strong coherence mostly for forests, with circa 1 year delay and – a weaker coherence – with circa 3 year delay. For pastures these interannual patterns are hardly recognizable. The strongest relationships occur between condition of the vegetation and T and ETo – they show high coherence in both forests and pastures. There is a significant cohesion with 8-16 month (ca. 1 year) delay and 20-32 month (ca. 2 year) delay. More time lagged significant correlations between vegetation indices and T occur for forests than for pastures, suggesting a significant lag in the forests' response to the changes in T. Yet, the surprisingly high coherence between vegetation condition and ETo of 2 year delay, which occurs for pastures along the whole study period is interesting, because low grassy vegetation seems to be unrelated to interannual weather conditions. To explain this, further, in-depth research is required, based on the increasingly longer research material. Additionally, as a division into just three vegetation types (broadleaved forest, coniferous forest, pastures) is rather coarse, a more detailed classification into e.g. tree species should be prepared, and the selected species-homogenous areas should be investigated in the future.

Another important methodological conclusion is the observed differences in forest NDVI and EVI, while for pastures NDVI and EVI values seem to be very similar. Finding out which of these indicators is more suitable for which type of forests and for low grassy vegetation requires separate, extended research, and should be conducted in the future.

The research presented in this study fills the knowledge gap on the coherence between vegetation condition and meteorological elements in the temperate zone. However, the obtained results might be useful for researchers working on this topic in other climatic zones. Identifying the climate-induced variability in the condition of vegetation is particularly important in the context of the recent climate change, and plants' impact on mitigation of the climate change.



Funding: This work was supported by the National Science Centre in Poland [grant number 2021/41/B/ST10/04113].

Data availability: The data that support the findings of this study were derived from the following resources available in the public domain: https://lpdaac.usgs.gov/products/; https://cds.climate.copernicus.eu/.

500    Author contribution: Kinga Kulesza: Conceptualization, Methodology, Formal analysis, Investigation, Data Curation, Writing - Original Draft, Writing - Review & Editing, Visualization. Agata Hościło: Conceptualization, Writing - Review & Editing, Supervision, Project administration, Funding acquisition.

The authors declare that they have no conflict of interest.



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
