# Peer review of "Coherency and time lag analyses between MODIS vegetation indices and climate across forest and grasslands in European temperate zone"

_EGUsphere, 2023_

## Referee Comment (RC2)

[referee-annotated manuscript omitted]

---

## Author Response (AR1)

REVIEWER 1

The manuscript presented a quite interesting research on the relationship between vegetation and climatic factors across different timescales using the Wavelet coherence method. In general, the manuscript is well structured, the results is clearly described and adequately connected previous research. I suggest to further improve the manuscript by clarifying some core elements and evaluating the robustness of the results.

Thank you for your appreciation.

1. L400: I suggest change "parameters" to "spectral bands"

   Yes, the term 'spectral bands' will suit better in this line.

2. L270-L275: The sentence "The high cohesion values indicate that the data sets exhibit high correlation in a year given in the x axis, and with a delay given in the y axis." Is unclear for me. I think a "delay" refer to one signal move lead/lag another but cannot be mixed with "correlation". I noticed that you seems mixed use the "delay" and "high coherence" many places in the manuscript. According to the section 2.5.2, the "delay" is the time-lag revealed by WT analysis. In other words, you cannot say there is a 1 year delay between T and NDVI when you see high (significant) coherence around 12 month scale (e.g. Fig.7 a) as the arrow in the figure indicate the two signals at yearly scale are almost in-phase (I.e. no-delay or very short delay). In this case, I suggest the delay and time-lag information should be further clarified.

   Yes, this may be a little confusing, but, to the best of my knowledge, wavelet coherence serves as a method to precisely combine the correlation and delays/lags. It can be used to determine whether significant wavelet spectra peaks observed at a given time in one signal, correspond with those observed by the other signals, as in the work of e.g. Mbatha and Xulu (2018). However if there are hardly no patches of significant wavelet spectrum determined in WT (and this is the case), the patches of high coherence in the WC scalogram might suggest that in such place there is a pattern in one signal which is similar to the pattern in the second signal, but delayed with a certain lag. Many papers (cited in the paper) use terms 'correlation' and 'delays/lags' in relation to wavelet coherence. In a similar way wavelet coherence results are presented in e.g. the paper of Ghaderpour et al. (2023) and Carl et al. (2013). Many papers also describe that right arrows suggest that series are completely in-phase, i.e. increases in signals are correlated, while left arrows suggest inverse correlation. The arrows, however, can be pointing any direction, suggesting that one signal leads/lags another, but if the arrows are pointing right it may suggest that both signals are in the same phase, or that one signal is leading/lagging by a full period under consideration. In this case, if we see a patch of a high coherence with the 12-month delay given in the y axis it means that the pattern we observe now in the meteorological variables is correlated to the pattern that we see 12 months later in NDVI. The signals are in-phase, because e.g. exceptionally high temperatures are followed by similar pattern of exceptionally high values of NDVI. If the delay/lag was shorter, then we would see a patch of high coherence in the same place on x axis, but shifted toward smaller values on the y axis (e.g. 6-months, or at the edge of the scalogram, which would suggest that the response of the vegetation is almost immediate – this is also the case in 2018, and this is described in lines 301-304). This way of interpreting these scalograms is, in my opinion, additionally proved by the

results of the Pearson's correlations with time lag. The similar information that we get from these two approaches is described in detail in many places in the paper (e.g. in lines 296-299: "Small areas of high positive correlation of circa 1 year delay between NDVI and NAO appear mostly for coniferous forest in the period 2013-2016 and 2018-2021, as well as for broadleaved forest in the period 2015-2016 and 2019-2020. This is additionally proven by the significant positive Pearson's correlation between NDVI and NAO for 11-month delay"). However, Pearson's correlation with time lag gives more general information about the correlation between two signals, because it is based on all years in question (equivalent of a patch in specific position on y-axis, but stretched on all years on x-axis on the WC scalogram). I hope this explanation justifies the use of the terms 'correlation' and 'delay/lag' as they are presented in the paper.

3.  The arguments between L449 and L454 is also quite confusion for me. Why the highly seasonal vegetation can prevent the detection of intra-annual interaction?

    Yes, maybe it was not clear enough. It does not prevent to detect the intraannual relationships, but they are much harder to detect, which is explicitly written in lines 449-450: "intraannual relationships, with the time lag smaller than 1 year are a bit harder to detect than similar relationships in the tropical or subtropical zones". It is because of the seasonality of vegetation in the warm temperate zone. This refers mostly to the Pearson's correlation with time lag, where all months, both spring and autumn/winter ones are correlated together. While severe weather condition occurring at the beginning of the growing season can induce poor vegetation condition in summer and autumn, the same severe condition occurring in autumn have much smaller influence on vegetation condition in winter (because the vegetation is very limited or paused, anyway). So if all months are correlated at once, the outcome might be weaker. Nevertheless, I changed slightly the text, so the statement is not so strong.

4.  The study investigated the cross-scale interaction relationship between vegetation and climatic signals over three common land types in Poland. It would be interesting to know how the relationship presented in Fig.7 and Fig. 8 vary within each land type. This would add robustness to the identified pattern at land type level.

    Yes, this is a good point, and we actually have already done some preliminary analyses. Please have a look at our previous paper (Kulesza K., Hościło A. 2023. Influence of climatic conditions on NDVI variability in forest in Poland (2002-2021), *Meteorological Applications*, 30(5). DOI: 10.1002/met.2156), in which the area of Poland is divided into 8 nature-forest lands, so we could have investigated the spatial distribution of vegetation condition. Here, we initially also applied the same division (into 8 spatial units). But, while we have three land types, and each of them should be spatially divided into 8 units, and all this should be done for NDVI and EVI, we would eventually have so much material to analyse and describe that it would be definitely too much for one paper (which is already quite extensive). So we decided not to include such results here. However, we plan to investigate such spatial distribution in next papers (it is a part of a bigger project). On the other hand, these preliminary results showed us that there are no big differences between individual parts of Poland. Poland is in fact a nine-largest country in Europe, but when it comes to response of different types of vegetation to changes in meteorological conditions it might be considered as almost homogenous, and therefore representative for whole European warm temperate zone. It is now stated in lines 461-466: "Finally, it should be noted that in this study all

pixels within the respective vegetation masks were spatially averaged, in order to produce single time series. However, the relationship between vegetation indices and meteorological elements may vary within each mask. Some initial, sample results (illustrated in Fig. A1 and A2 in Appendix A) showed us that there are no big differences between individual parts of Poland. Poland is in fact a nine-largest country in Europe, but when it comes to response of different types of vegetation to changes in meteorological conditions it might be considered as relatively homogenous, and therefore representative for whole European warm temperate zone". However, as we plan to further research the variation within each land type, I added this information in the Conclusion section (lines 490-491: "The species-homogenous areas might also be further divided spatially, in order to check in detail the differences in species responses to the changes in meteorological conditions in different regions of the study area.").

5. To reveal the interaction relationship between vegetation and climate factors across multiple timescales, the spectral analysis techniques in frequency domain such as Fourier analysis, cross-spectral analysis, as well as Wavelet coherence were widely applied in many studies. It's better to evaluate the results of this study broader by comparing other relevant studies, especially those using spectral analysis in frequency domain.

Yes, this is an interesting point. In fact, Fourier analysis is well established and has often been used in environmental research so far, but its great disadvantage comes from the fact that 'it presents only resolution on frequency and not in time' (Moreira, Fontana and Kuplich 2019). That is why we focused on wavelets mostly. However, regarding WT, we discussed cycles found by wavelet analysis in meteorological elements (lines 362-365: "Similarly, no significant interannual cycles in meteorological time series in central Europe were found in other works, using much longer time periods, regarding T (…), P (…) and NAO (…)"). Similarly we did with the most important in this paper, i.e. WC results. When it comes to wavelet coherence and the relationships between vegetation condition and meteorological elements and teleconnection indices, we made a broad research throughout the papers worldwide, and found out that there is not many papers dealing with this topic. I would dare to say that most of similar papers were discussed, even if their study areas were quite remote from Poland (e.g. Brasil, Africa). I could, however, extend the discussion to cover some more interesting papers, if you have anything specific in mind.

REVIEWER 2

Thank you for the submitted manuscript. It is potentially interesting but needs revision. The submitted study analyzes the statistical relationship between optical vegetation indices (NDVI and EVI) and meteorological variables (temperature, precipitation, evapotranspiration, vapor pressure deficit) as well as teleconnection indices (NAO and NCP). The study uses Poland as a study area and uses wavelet transforms as the main analysis method. I believe that while the methods are not highly novel, they are applied correctly, and the analysis is carried out well. I think that to some extent the presented results do not fully support the conclusions in the manuscript. The interpretation of results, in my opinion, seems often based on visual interpretation of the figures, and was not fully clear to me.

Thank you for your words of appreciation. Yet, I do not understand clearly the statement that interpretation of results is based on visual interpretation of the figures. Yes, in the first part,

where the anomalies are being described and discussed (mostly Fig. 3) we visually find the big anomalies, and find confirmation of the big anomalies in the literature describing drought events. This part of the paper is indeed more 'qualitative'. However the second (main) part of the paper, with the results of the WT and WC, is for sure 'quantitative'. We interpret the WT and WC scalograms, but we refer to the patches that are statistically significant (circled with black thick line) and inside the COI. Also the Pearson's correlation with time lag is assessed in terms of its statistical significance, and only the significant values are interpreted.

Furthermore, I believe that discussion of the results focuses too much on the results of other studies, rather than exploring the presented ones. This is why I would recommend major revisions.

According to this remark, and many comments made in the PDF, I changed the discussion, especially removed some unnecessary referencing to other studies. I hope that you will find these changes satisfactory.

General comments:

1. The analysis averages all data (vegetation indices and climatic variables) into a single time series (for three landcover types), thus averaging across quite a big area. Can you give some background on why you decided to do so and what the potential implications are? Poland is quite a big country, so I guess the relationship between vegetation indices and climatic variables is different across space. Inner Poland might be more continental than areas closer to the sea. Many extreme events you discuss throughout your manuscript likely have had different effects in the different areas of Poland. Having a spatial component might improve the conclusiveness of your results.

   Yes, of course, it is a very good remark. Yet surprisingly, we have already faced also contrary comments suggesting that Poland is not big enough to study only it. Not to mention dividing Poland into even smaller spatial units. However, we actually have already done some preliminary analyses in such smaller spatial units. Please have a look at our previous paper (Kulesza K., Hościło A. 2023. Influence of climatic conditions on NDVI variability in forest in Poland (2002-2021), *Meteorological Applications*, 30(5). DOI: 10.1002/met.2156), in which the area of Poland is divided into 8 nature-forest lands, so we could have investigated the spatial distribution of vegetation condition. Here, we initially also applied the same division (into 8 spatial units). But, while we have three land types, and each of them should be spatially divided into 8 units, and all this should be done for NDVI and EVI, we would eventually have so much material to analyse and describe that it would be definitely too much for one paper (which is already quite extensive). So we decided not to include such results here. However, we plan to investigate such spatial distribution in next papers (it is a part of a bigger project). On the other hand, these preliminary results showed us that there are no big differences between individual parts of Poland. Poland is in fact a nine-largest country in Europe, but when it comes to response of different types of vegetation to changes in meteorological conditions it might be considered as almost homogenous, and therefore representative for whole European warm temperate zone. It is now stated in lines 461-466: "Finally, it should be noted that in this study all pixels within the respective vegetation masks were spatially averaged, in order to produce single time series. However, the relationship between vegetation indices and meteorological elements may vary within each mask. Some initial, sample results (illustrated in Fig. A1 and A2 in

Appendix A) showed us that there are no big differences between individual parts of Poland. Poland is in fact a nine-largest country in Europe, but when it comes to response of different types of vegetation to changes in meteorological conditions it might be considered as relatively homogenous, and therefore representative for whole European warm temperate zone".

2.  Fig. 3 is important for the whole manuscript as the anomalies identify potential events. The time axis in the figure, as well as the grey/white shade, might have confused me. Do the tick marks (e.g., 2014) relate to the beginning of the year (2014) or the middle of the year? If they refer to the beginning of the year the grey and white shading is confusing me. With so many subplots it would be best to have grey and white shading indicate one year exactly.

    Thank you for drawing my attention to this. If you look closer to the beginning of the x axis in the Fig. 3, you will see that it starts with a tick and the '2002' – so the tick mark refers to the beginning of the year. Then, moving to the right, ticks appear every two years (next tick is for the beginning of 2004). In between we have two years then: 2002 and 2003, and growing season of each of them is shaded with grey colour (although there is a mistake in the caption: grey-shaded are months from April to October, and NOT April-September, as it is written in the caption, I changed it). We preferred to grey-shade the growing seasons only, so the reader could easier find the months of the warm half-year (in this case you can find value for August counting from first grey-shaded month, i.e. April, instead of counting from January). However, to make this graph more clear I added ticks for each year (instead of every two years, as it is now). I believe that keeping the grey-shading only for warm half-year (months April-October) is advantageous, and I would prefer to keep it this way.

3.  Many of the wavelet spectra in Fig. 4 seem to have a significant high power area around 2010 and 1 year period. Also, the time series in Fig. 3 have very negative NDVI anomalies in 2010. At several points in the manuscript (e.g., line 475) it is said "Thanks to conducting the WT analysis, no significant intra- or interannual cycles were detected in both vegetation (NDVI and EVI) and meteorological variables." Does this refer to the spectra only showing a blob for one year (2010) rather than a line with high power across all years? It is not clear to me why these patterns are disregarded and not investigated.

    Thank you for this remark, indeed this should be improved to better express the actual meaning. There are two things here to explain. First is the patch of the high power of the wavelet spectrum that occurs around 2010, with a cycle of approx.. 1 year. Indeed, it is visible mostly for pastures, for both NDVI and EVI anomalies, and to some extent for both forests for NDVI. It comes from big negative NDVI/EVI anomalies in January 2010 and December 2010. Both months had exceptionally big snow cover, which resulted in very low values of NDVI and EVI. As a result, a signal of NDVI/EVI in 2010 presents a 'sinusoid', which was indeed detected by the WT. The existence of this patches was noticed in lines 255-258: "The pulse of a half-year and 1 year cycle of fluctuations in NDVI is marked around the 2010 for all three types of vegetation (…). Although they are statistically significant, neither the power spectrum is strong, nor they last long. The EVI shows similar pattern for pastures, but much fewer statistically significant fluctuations for broadleaved and coniferous forests (…)", but without an explanation from where it come from. And this short explanation I will add in

subsequent lines 258-260 ("These pulses come from the big negative NDVI and EVI anomalies in January and December 2010, caused by extensive and persistent snow cover that significantly changed the values of spectral reflectance"). The second thing that should be explained here is the reason why we computed the WT at all. In order to investigate the relationships between meteorological variables and vegetation condition, and the delays/lags in spectral response of vegetation to the triggering meteorological factors, we had to make sure that the high coherence observed between two signals in WC scalograms is not incidental. If NDVI/EVI or meteorological elements would show some natural cyclicality that would be more or less stable in the whole time period (e.g. 3-year-long cycles), and later WC would suggest a 3-year cohesion, that could have been a coincidence. However, as there are no stable-over-whole-period cyclical components in both NDVI/EVI and meteorological elements, then presumably the spectral response of the vegetation to the triggering meteorological factors is caused rather by the actual relationship between these variables, than is the result of a coincidence (it is written in lines 367-368). So yes, you are right, writing that "this [no significant intra- or interannual cycles were detected in both vegetation and meteorological variables] refers to the spectra only showing a blob for one year (2010) rather than a line with high power across all years". But to make it more clear and understandable I improved this. Now e.g. in lines 470-471 (and similar places in the text) it is said: "Thanks to conducting the WT analysis, no significant and stable over the whole time period intra- or interannual cycles were detected in both vegetation (NDVI and EVI) and meteorological variables".

4. I believe the discussion of the results is lacking in several ways. First there is no real discussion of why the chosen landcover types might behave differently. This makes me ask why they were chosen in the first place, especially as other very relevant landcover types like croplands are not analyzed.

As the main goal of this study is to analyse the delays/lags in the response of vegetation condition to triggering meteorological factors, we chose two different vegetation types to see if they would react differently to the same weather conditions. The assumption was that one type reacts quickly (pastures), while the other slowly (forest). The part of the discussion in which we highlight the differences between forest and pastures is in lines 451-455 ("For instance, the NDVI signal coming from forest reflects not only the trees vigour, shaped by weather conditions, but also the "noise" from the understorey and other effects like pests, herbivores, pathogens and forest management. Grassland seem to be mostly free from such problems. Its response to the triggering meteorological factor is usually quick. Also, the quality of grassy vegetation in one year does not have a major effect on its quality in the subsequent year.") and 383-387 ("Indeed, forest vegetation response to water deficit can be delayed, especially depending on the tree type (broadleaved or coniferous) and species. (…) the surprisingly high coherence of 2 year delay, which occurs for pastures along the whole study period is interesting, because one would expect current low grassy vegetation to be independent of the weather conditions in the previous growing seasons."). At the same time, we purposely excluded croplands, because their spectral reflectance changes many times in the year, regardless of the weather (e.g. ploughing and harvesting couple times in year), which would make the results very hard to interpret. For the same reason we excluded also some classes of grassland, which is described in lines 455-458.

5. Some of the references used for pastures, study areas in Brazil or Iran (for example). I think this is too far of a stretch without presenting evidence that they are comparable to pastures in Poland. I think this could be improved to make the manuscript more relevant for Biogeosciences readership.

   Yes, I am aware that papers discussed here are from remote study areas (Brasil, Africa, etc.). However, these were truly the only papers that I could find that dealt with similar topic (relationship between vegetation condition and weather conditions and delays/lags) and used similar (WC) methods. I know that the pastures in Brasil are incomparable with European ones, but in this case, the aim was to present the results from other places, to show how much they are different from the European ones, and to show that – in general – there is a connection between temperature (or precipitation) and vegetation (though it is much shorter delay). In places where it was possible, we discuss the European papers (Germany and Italy).

6. Second, a large part of the discussion focuses on individual disturbance events, mostly large drought events. This discussion is mostly based on other studies, and I think the link to the presented results is weak. I struggle to find some of the drought events in Fig. 3, as for example negative anomalies of precipitation, NDVI or EVO or positive anomaly of temperature.

   Yes, when we discuss drought events, many other studies is discussed. However, I do not agree that the link to the presented results is weak. For each drought event we introduce some other papers and then describe how big where the anomalies of meteorological elements/teleconnection patterns and vegetation condition, according to our research results. For 2003 in lines 339 and 343, for 2015 in lines: 343-345, for 2018 and 2019 in lines: 349-353 and 355-356. When it comes to Fig. 3 I hope that after my clarification and changes made to this figure, now you will find it easier to localise specific anomalies.

7. A 2010 drought where Christian et al. 2010 is referenced, did never really extend to Poland, I am wondering why it is discussed. Overall, the link between results and discussion must be stronger, this is my main point. 2010 seems like an anomalous year, even visible in the power spectra in Fig. 4. I believe a major flood happened in Poland during this year. See https://en.wikipedia.org/wiki/2010_Central_European_floods or Pińskwar et al. 2019: Observed changes in extreme precipitation in Poland: 1991–2015 versus 1961–1990. I don't understand how this was missed in the discussion of the results especially as it seems to be visible in Figs. 3 and 4., while other discussed droughts are not visible to me, unless I am misinterpreting the figures. The discussion needs to be more comprehensive and explain why forest and pasture classes might (or might not) behave differently in the analysis.

   I agree that discussing e.g. 2010 drought, which had no impact on the vegetation in Poland might be pointless, so I removed the whole paragraph describing European drought events that did not happen in Poland. I hope this (together with other improvements) would satisfy your remark about too extensive discussing of other studies. When the flood from 2010 is concerned, indeed the big positive anomalies of P occurred in May, August and September, but no big anomalies occurred in the growing season of 2010 for both NDVI and EVI (Fig. 3). In turn, Fig. 4 shows the pulse of cyclicality in 2010, but its reason I already explained (extensive and long-lasting snow

cover in January and December). As the paper mostly focuses on vegetation, I think discussing additionally the 2010 flood might be too much.

8. Is there a possibility that the ENSO cycle is affecting the dynamics of NAO and other variables? ENSO has a period of 3-4 years in many cases, it would add an interesting layer of content to discuss. Here are some potential references discussing links between ENSO and NAO.

Thank you very much for finding these references. It is very probable that ENSO affects NAO, as atmosphere is a system of interconnected vessels. However, to prove if ENSO indirectly affects vegetation in Europe, separate study is needed. So, in order to not expand the discussion too much, I only mentioned such possibility in lines 435-437: "As the atmosphere is a system of interlinked vessels, the NAO may itself be influenced by other teleconnection systems, e.g. 3-4-year cycle of ENSO (King, 2023). Thus the indirect effect of ENSO on vegetation condition in Europe might be investigated in the future".

REMARKS FROM EDITOR:

Thank you for your submission to Biogeosciences. I agree with the reviewers that the study presents an interesting analysis with a focus on Poland.

Thank you for your appreciation.

1. Both reviewers brought up issues related to wanting to know more about variations within the vegetation types. While I agree with the authors that this would create a lot of work, since two independent reviewers asked about this and anticipate other readers would be curious, it would be worth adding a supplemental figure that looks at some differences within the three vegetation types (potentially picking one of the types and looking at differences within the group across the country) so that the reader can see how much Figs. 6-8 change within a given vegetation type. Or comment on this supplemental analysis (not shown) about the homogeneity of responses within the vegetation type, and thus how much these might change if different subsections of the country are analyzed.

   Following the editor's suggestion, we added the Appendix A, which contains two figures (Fig. A1 shows the division into 8 nature-forest lands, while Fig. A2 shows the wavelet coherence for broadleaved forest in 3 selected, very different lands – we present only 3 lands because of the size restrictions). In the main text there is now a paragraph in the Discussion that refers to these preliminary, sample results in Appendix A (lines: 461-466). I am not sure, whether it should be an Appendix or a Supplement. I leave this decision to the technical editor.

2. Some additional points to consider when I was reading through include: why was radiation not considered as a driving factor given its large impact on vegetation in the midlatitudes? I also recommend Fig. 5 and Fig. 6 y axes to be the same or comment on why they are not.

Thank you for noticing that Figs. 4 and 5 have different y axes than Figs. 6 and 7. I changed it, so now all y axes are scaled in months. When it comes to solar radiation, according to the literature review, solar radiation is a meteorological element which has smaller impact on vegetation condition than e.g. temperature. At the same time, solar radiation is highly correlated with temperature. Eventually, we decided to use only those elements which "are generally known to have a significant impact on the dynamics of vegetation productivity" (lines 134-135). However, in future studies it might be beneficial to include also solar radiation and its influence on vegetation.

3. I ultimately think the authors' responses to reviewers are comprehensive. We invite them to revise the manuscript. A final note that it is unclear how much authors plan to update their text/analysis based on their responses and some disagreements. In the revised manuscript, please either make an update to the manuscript or provide a detailed response about why no change would be made.

We have revised the manuscript in accordance with all suggestions of both reviewers and the editor. All changes are easy to track (track changes mode). Additionally we provide the detailed answers to reviewers' and editor's remarks, with updated line numbers, referring to the revised version of the manuscript (line numbers refer to 'manuscript_revised.pdf').

---

## Author Response (AR2)

REVIEWER 1

After reviewing all the previous comments and responses, I find myself mostly in agreement with the arguments presented. However, I have reservations about the handling of the concepts in the initial comments of review 1. Specifically, the authors' explanation of the wavelet coherence scalogram seems to conflate the terms 'delay' and 'timescale'. During their discussion of wavelet coherence scalogram, they appeared to mistakenly interchange the terms 'delay' and 'timescale'. For instance, in their response to the comments, they asserted that" ... if we see a patch of a high coherence with the 12-month delay given in the y axis it means that the pattern we observe now in the meteorological variables is correlated to the pattern that we see 12 months later in NDVI. The signals are in-phase, because e.g. exceptionally high temperatures are followed by similar pattern of exceptionally high values of NDVI. If the delay/lag was shorter, then we would see a patch of high coherence in the same place on x axis, but shifted toward smaller values on the y axis (e.g. 6-months, or at the edge of the scalogram, which would suggest that the response of the vegetation is almost immediate – this is also the case in 2018, and this is described in lines 301-304)...". In reality, the y-axis of the wavelet coherence scalogram signifies the timescale of synchronism between two signals, not the delay. The actual delay (in months) of the two signals at a specific timescale can only be determined by combining the phase difference (or phase-delay) indicated by the arrow direction (in radians) and the timescale. Upon reviewing the references mentioned in the response, namely Mbatha and Xulu (2018) and Ghaderpour et al. (2023), I found no indication attributing the y-axis of the scalogram as a delay in these papers.

In the revised manuscript, the authors argued in Lines 276-278 that "The high cohesion values indicate that the data sets exhibit high correlation in a year given on the x-axis, with a delay indicated on the y-axis." I am inclined to disagree with the assertion that the delay is depicted on the y-axis.

In these case, I sugget the manuscript should by further impoved before acceptance for publication.

While I still believe that my explanation to the initial comment no. 1 is valid and justified, I agree with the Reviewer to some extent. Hence, I removed this quoted sentence (lines 277-278) from the manuscript. I also made some further adjustments, in order to fully comply with the Reviewer's comment.

REVIEWER 2

Thank you for making substantial edits to the manuscript. The addition of the new supplementary figures and the adjustment of the axis in Fig. 3 are great. I believe the changes to the text have improved the manuscript and made the presented research clearer. Overall, I have no major points left that I would like to see changed in the manuscript. I leave it up to you whether you want to include the following minor points, I believe they would further improve the manuscript.

Remaining minor points

I would suggest another potential figure adjustment, that I previously missed. It might be beneficial for the reader to have the same y-axis scaling for both NDVI and EVI in Fig. 8. This would further enhance the idea that the correlation coefficients with NDVI, especially for time lags smaller than 5 are higher than the ones for EVI.

Yes, it might be beneficial, I changed accordingly.

I still believe some of the discussion related to very different ecosystems might be confusing rather than helpful. For example, around line 420, discussion relative to very specific vegetation in Brazil. I think the manuscript would be completely fine without it.

Yes, I removed this part describing the Caatinga vegetation (lines 419-421), as it is – indeed – referring to the specific vegetation type in Brazil. However, there are still lines referring to general vegetation patterns in Brazil and South Africa (lines 381-384, and 394-395). I believe they should stay.

EDITOR'S COMMENT

The reviewers found the responses to be sufficient, especially with regard to the major comments. Both reviewers have remaining comments that I agree all should be addressed (especially about the discussion of other biomes not included in the study and the difference between phase and coherence in the wavelet analysis) before the manuscript is published.

I made all necessary changes. I hope you will find the manuscript suitable for publication now. Thank you.